# Gender differences in motor and non-motor symptoms in individuals with mild-moderate Parkinson's disease

**Amit Abraham**[1,2], **Allison A. Bay**[3], **Liang Ni**[3], **Nicole Schindler**[4], **Eeshani Singh**[4], **Ella Leeth**[3], **Ariyana Bozorg**[3,5], **Ariel R. Hart**[3], **Madeleine E. Hackney**[3,5,6,7,8]*

**1** Department of Physical Therapy, Faculty of Health Sciences, Ariel University, Ariel, Israel, **2** Navigation and Accessibility Research Center of Ariel University (NARCA), Ariel University, Ariel, Israel, **3** Division of Geriatrics and Gerontology, Department of Medicine, Emory University School of Medicine, Atlanta, GA, United States of America, **4** College of Arts and Sciences, Emory University, Atlanta, GA, United States of America, **5** Rehabilitation R&D Center Atlanta Veterans Affairs Health Care System, Decatur, Georgia, United States of America, **6** Department of Rehabilitation Medicine, Emory University School of Medicine, Atlanta, GA, United States of America, **7** Emory School of Nursing, Atlanta, GA, United States of America, **8** Birmingham/Atlanta VA Geriatric Research Education and Clinical Center, Decatur, Georgia, United States of America

* mehackn@emory.edu, madeleine.hackney@gmail.com

**Data Availability Statement:** All relevant data are within the paper and its Supporting Information files.

## Abstract

### Background

Parkinson's disease (PD) affects both men and women with documented gender differences across functional domains, with findings varying among reports. Knowledge regarding gender differences in PD for different geographic locations is important for further understanding of the disease and for developing personalized gender-specific PD assessment tools and therapies.

### Objective

This study aimed to examine gender differences in PD-related motor, motor-cognitive, cognitive, and psychosocial function in people with PD from the southern United States (US).

### Methods

199 (127 men and 72 women; *M* age: 69.08±8.94) individuals with mild-moderate idiopathic PD (Hoehn &Yahr (H&Y) Median = 2, stages I-III) from a large metro area in the southeastern US were included in this retrospective, cross-sectional study. Motor, motor-cognitive, cognitive, and psychosocial data were obtained using standardized and validated clinical tests. Univariate analyses were performed, adjusting for age and housing type.

### Results

After adjustment for age, housing, PD duration and fall rate, men exhibited statistically significantly greater motor (Movement Disorders Society (MDS)-Unified Parkinson Disease Rating Scale (UPDRS)-II) and non-motor (MDS-UPDRS-I) impact of PD, and more severe

**Funding:** This work was supported by the United States Department of Veterans Affairs (Award: 5IK2RX000870-06) awarded to Madeleine Hackney, the National Center for Advancing Translational Sciences of the National Institutes of Health (Award: UL1TR002378), the Emory University Udall Center (Award: P50NS071669-04) and the Parkinson Foundation (Award: A1-2016). The funders had no role in study design, data collection and analysis, decision to publish, or preparation of the manuscript.

**Competing interests:** The authors have declared that no competing interests exist.

motor signs (MDS-UPDRS-III). Men exhibited worse PD-specific health-related quality of life related to mobility, activities of daily living, emotional well-being, cognitive impairment, communication, and more depressive symptoms. Men performed worse on a subtraction working memory task. Women had slower fast gait speed.

## Conclusions

In the southeastern United States, men may experience worse PD-related quality of life and more depression than women. Many non-motor and motor variables that are not PD specific show no differences between genders in this cohort. These findings can contribute to the development of gender-sensitive assessment and rehabilitation policies and protocols for people with PD.

## Introduction

Parkinson's disease (PD), the second most common neurodegenerative disease, results in motor and non-motor (i.e., cognitive, sensory, etc.) impairments that deteriorate quality of life (QoL) and wellbeing [1]. PD burdens men and women affected by the disease as well as families, health care systems, and societies [2]. Estimated costs associated with PD are as high as $34 billion per year [3]. PD affects about 1% of individuals over 60 years of age [4], and more than 10 million men and women worldwide, including 1,238,000 Americans by the year 2030 [5].

Differences between men and women in PD epidemiology, clinical presentation, and response to pharmacological therapy have been reported [6–8]. These differences may be explained by gender-specific age-related physical and cognitive changes, role expectations, societal attitudes [9], gender differences in life expectancy [10] and age at PD onset [7] (women are older at onset). Additional factors, such as social support [11] and living environment [12], also impact functional and psychological status and quality of life (QoL) in individuals with PD.

Indeed, there may be some pathophysiological basis for differences in several motor, cognitive and psychosocial factors between men and women with PD. Gender seems to influence the expression of several polymorphisms in PD. Genetic factors might differentially influence the manifestations of PD in men and women. It is important to consider estrogen and reproductive factors also. Female and male reproductive hormones have important influence that can significantly alter individual thermoregulatory responses during the lifespan [13]. Vascular responsiveness exhibits age- and sex-based differences in healthy subjects and trauma patients and estrogen appears to be protective [14]. Higher estrogen and progesterone levels may be related inversely to upper gastrointestinal cancers, which may help explain lower incidence rates of such cancer in women compared with men [15].

There may be a positive effect of estrogens on the dopaminergic system. Gonadal hormones and sex chromosomes might modulate PD risk by influencing epigenetic mechanisms. Preclinical evidence suggests potential neuroprotective effects of estrogens against dopaminergic damage through anti-inflammatory, anti-oxidative, and anti-apoptotic mechanisms in addition to possible inhibitory effects on the formation and stabilization of $\alpha$-synuclein fibrils—a pathological feature of PD [16]. Women with presumed higher cumulative lifetime levels of both endogenous and exogenous estrogen had a significantly reduced risk of PD relative to

those with lower lifetime estrogen exposure [17]. Further, neuroimaging findings support structural and functional signatures in women with PD, that are characterized by a more preserved presynaptic system and higher striatal dopaminergic levels at disease onset compared with men [16].

In addition, sociocultural and demographic factors may affect men and women differently in the southeastern United States. Older adults in the Southeastern United States, compared to the rest of the country have considerable health and health-related quality of life (QoL) challenges [18]. The region has high numbers of individuals with other debilitating chronic diseases besides PD, such as diabetes mellitus, cardiovascular disease, obesity and cancer. For example, diabetes has a great influence on life expectancy for many older adults who live in the south, and persons with diabetes born in the south were more likely to have developed chronic conditions or disabilities and spent more of their life with the chronic conditions compared to other regions in the United States [19]. In patients with other comorbid conditions (e.g., HIV), those in the South were on average more vulnerable to falls than were individuals from other regions [20]. As regards potential environmental risk factors for PD, women tend to have a lower exposure to occupational toxins and a lower incidence of head trauma than men, reflecting differences in behavioral and social factors [16]. Examining the status of people with PD from this region is important and whether gender modifies this relationship deserves more research.

The role of gender in PD, motor, and motor-cognitive outcomes is not entirely clear to date, with conflicting findings across studies [21]. For example, while some studies found activities of daily living (ADL), cognition and communication-related aspects of health related QoL (HRQoL) were rated lesser in men [22, 23], other studies reported worse disability, QoL and HRQoL in women [24, 25]. Further, studies that directly compare men and women on performance of frequently used assessments of functional mobility and motor-cognitive function are rare, although these data and comparisons would be of great interest to the rehabilitation field. Empirical data regarding the influence of gender on PD management and care is scarce [26], especially regarding how gender impacts response to non-invasive and non-pharmacological therapies, which should influence the development and implementation of personalized rehabilitative protocols for individuals with PD [6, 26]. Therefore, further exploration of the impact of gender on PD-related motor and non-motor symptoms as well as performance on standard clinical measures of mobility, motor-cognitive and cognitive function is warranted and could promote the development of PD patient-tailored, gender-sensitive rehabilitation assessment and therapies. The following example illustrates this issue's importance. A study of 1,463 participants (914 men with a mean age of 64.5±10.37 and 549 women with a mean age of 65.7±10.97) found at diagnosis of PD, women were more likely to be less educated, more anxious, and faced greater instrumental activities of daily living (IADL) disability [25]. Yet, current physical and rehabilitative therapies for PD do not incorporate this important information into clinical decision-making, rendering gender-sensitive treatment protocols effectively unavailable to date [26].

Gaining additional knowledge regarding the role of gender on PD, and particularly within the Southeastern United States region is important for: 1) further understanding the impact of PD on both men and women from a comparatively more vulnerable population of older adults (Southern U.S. older adults) to better interpret findings; 2) gaining a window into future research about disease mechanisms and treatment strategies; and 3) promoting gender-specific PD rehabilitation and therapies. Addressing the importance and need for additional research into "detailed and specific cognitive and functional assessment by gender," [27] this paper will examine gender differences in cognitive and functional variables of rehabilitative interest. The terms "sex" and "gender" are not equivalent but are frequently used interchangeably in the

literature [6], with "gender" including both biological (i.e., sex) as well as social-environmental, cultural, and personal aspects and implications of being a woman or a man (i.e., gender) [6, 28, 29]. Because all variables considered in this study are behavioral, which encompasses social components, and given that "gender" is viewed to be more relevant for describing how biological and social components affect health outcomes [29], this paper uses the term, gender, which is in line with PD literature [6, 9, 28, 30–32]. This study sought to examine and compare performance in a sample of men and women with diagnosed mild-moderate PD (Hoehn & Yahr stages I-III) residing in the Southeast of the United States in motor, motor-cognitive, cognitive, psychosocial and PD-specific function.

## Methods

The study was approved by the Emory Institutional Review Board and the VA Review committee (Protocols IRB060613, IRB055977, and IRB047231). All participants gave written informed consent prior to participation in study activities.

### Participants

Data from 199 (127 men and 72 women; *M* age: 69.08±8.94) individuals with idiopathic PD (H&Y Median = 2, stages I-III) were included in this retrospective, cross-sectional study. To be included in the study, all participants had to have a diagnosis of idiopathic "definite PD" that was determined by a board certified Movement Disorders trained neurologist [33], which meant that at time of diagnosis they reported unilateral onset of symptoms, they exhibited three of the four cardinal signs of PD (i.e., rigidity, tremor, bradykinesia, and postural instability) and exhibited clear symptomatic benefit from antiparkinsonian medications [34]. Inclusion criteria were ability to walk ten or more feet with or without an assistive device and having no other diagnosed neurological disorders. Exclusion criteria were major psychiatric illness, history of stroke, or traumatic brain injury, alcohol abuse and/or use of antipsychotics, severe cardiac disease, and any other significant co-morbid disease that would impair ability to participate. Participant demographics are detailed in Table 1.

### Data collection & outcome measures

Data were collected using surveys, which were self-completed at home on paper, online, in person at the assessment site or over the phone with research assistant help. Research assistants verified that participants understood the intent of questionnaires. Participants were asked to verify that they self-completed the surveys. Motor and cognitive data were collected on-site. Standardized, valid, and reliable clinical measures were used for assessing motor, cognitive, and psychosocial function.

### Socio-demographics and disease severity & stage

Socio-demographic factors (age, race, education, number of falls in the past year, number of comorbidities, number of medications, years with PD, housing, transportation, frequency of leaving the house, freezer status (i.e., freezing of gait: yes or no), and use of assistive device) were collected via a health questionnaire. In addition, The Composite Physical Function (CPF) questionnaire, which characterizes participants' ability to complete activities of daily living (ADLs) (/24; higher = better), was completed [35]. Participants' self-reported frequency and duration of physical activity was assessed using the Physical Activity Scale for the Elderly (PASE) questionnaire, which has scores that range from 0−400, with higher scores indicating

**Table 1. Participants' demographics and clinical characteristics.**

| | Total (n = 199) | Men (n = 127) | Women (n = 72) | Difference (*P* value) |
|---|---|---|---|---|
| Age (y) | 69.08±8.94 | 68.17±9.44 | 70.68±7.79 | 0.057 |
| BMI | 26.40±4.83 | 25.05±5.50 | 27.16±4.25 | 0.006** |
| Race | | | | 0.109 |
| Black | 30 (15.2) | 14 (11.1) | 16 (22.2) | |
| White | 155 (78.3) | 103 (81.7) | 52 (72.2) | |
| Other | 13 (6.6) | 9 (7.1) | 4 (5.6) | |
| Hoehn & Yahr[a] | 2 (1) | 2 (0.5) | 2 (1) | 0.977 |
| MoCA (/30) | 25.18±3.9 | 24.91±4.2 | 25.67±3.4 | 0.223 |
| Education (years) | 16.36±2.29 | 16.44±2.26 | 16.22±2.34 | 0.530 |
| Number of Falls in the Past Year | 7.76±37.73 | 6.9±33.36 | 9.26±44.58 | 0.672 |
| Number of Comorbidities | 3.42±1.81 | 3.24±1.79 | 3.72±1.82 | 0.074 |
| Composite Physical Function (/24) | 19±5.08 | 19.35±4.99 | 18.39±5.2 | 0.201 |
| Physical Activity Scale for the Elderly (PASE) | 103.34±70.2 | 104.17±69.4 | 101.95±72.2 | 0.843 |
| Number of Medication | 5.9±4.13 | 6.15±4.26 | 5.51±3.93 | 0.336 |
| Years with PD (y) | 6.64±4.58 | 6.34±4.49 | 7.16±4.73 | 0.235 |
| Housing | | | | 0.005** |
| Assisted/senior living | 21 (10.7) | 7 (5.6) | 14 (19.4) | |
| Self/independently | 176 (89.3) | 118 (94.4) | 58 (80.6) | |
| Transportation | | | | 0.089 |
| Family | 40 (20.3) | 30 (24) | 10 (13.9) | |
| Public | 6 (3) | 2 (1.6) | 4 (5.6) | |
| Self | 150 (76.1) | 93 (74.4) | 57 (79.2) | |
| Service | 1 (0.5) | 0 (0) | 1 (1.4) | |
| Leaving House Frequency | | | | 0.909 |
| 0–2 per week | 16 (8.2) | 11 (8.8) | 5 (7) | |
| 3–4 per week | 55 (28.1) | 35 (28) | 20 (28.2) | |
| Daily | 125 (63.8) | 79 (63.2) | 46 (64.8) | |
| Freezer[b] | | | | 0.902 |
| No | 50 (62.5) | 33 (61.1) | 17 (65.4) | |
| Yes | 30 (37.5) | 21 (38.9) | 9 (34.6) | |
| Use of Assistive Device | | | | 0.859 |
| No | 118 (69.8) | 73 (68.9) | 45 (71.4) | |
| Yes | 51 (30.2) | 33 (31.1) | 18 (28.6) | |

Values are presented as Mean ± SD for continuous variables, and n (%) for categorical variables.

BMI = Body Mass Index.

*P* values were calculated with t-test/ANOVA for continuous variables and Chi-square test for categorical variables.

[a] Values are median (Interquartile range). *P* value obtained from Mann-Whitney U test.

[b] Freezer = experiencing freezing of gait.

*$p < 0.05$

**$p < 0.01$

greater physical activity [36]. The MoCA, which assesses global cognition through 8 cognitive domains [37], was administered. A higher score represents better global cognition.

Measures of disease severity included the MDS-UPDRS parts I-IV, on which higher scores indicate greater disease severity [38]. Part I assesses the non-motor impact of PD on patients' experiences of daily living, including apathy, anxiety, sleep, hallucinations, fatigue etc. Part II assesses the motor impact of PD on the patient's experience of daily living. Items covered

include drooling, eating, handwriting, and rolling over in a bed. Part III assesses the motor signs of PD. This section is administered by a trained examiner wherein the examiner observes the participant performing several motor tasks (i.e., rising from a chair, toe tapping, etc.) The examiner also observes tremor, muscle rigidity, posture, and gait. Part IV is delivered in interview format and is a self-reported indication of medication-related motor fluctuations (MRMF): dyskinesias, time spent in the off state, functional impact and complexity of fluctuation, and off-state dystonia. Current Hoehn & Yahr (HY; Stages 1–5) staging at the time of the assessment was also measured by the MDS-UPDRS [38].

PD quality of life was assessed using the self-report measure, the Parkinson's Disease Questionnaire 39 (PDQ-39). The PDQ-39 measures health-related QoL over the past month for PD patients in 8 domains: mobility, activities of daily living (ADLs), emotional well-being, stigma, social support, cognition, communication, and bodily discomfort, plus a summary index (PDQ-39 SI) which indicates the global impact of PD on health status, which is internally reliable and valid [39–41]. Lower scores indicate better quality of life.

## Motor

Fullerton Advanced Balance (FAB; /40) was used to measure static and dynamic balance and is valid for use in PD [42]. Lower FAB scores indicate difficulty with higher level static and dynamic balance tasks. Gait speeds (preferred, backward, fast) were obtained by recording the time (in seconds) for the participant to walk 6 meters at their self-selected ('normal') gait speed, as fast as possible ('fast') forward gait speed, and self-selected backward ('backward') gait speed, and are reported in meters per second (m/s). Number of steps required to complete each task was also recorded. Backward walking is more likely to be impaired earlier in the disease than forward walking and is more sensitive to change over time [43]. In the Timed Up and Go (TUG) test, individuals are instructed to stand up from a chair, as quickly and as safely as possible, walk three meters, cross a line marked on the floor, turn around, walk back, and sit down. The TUG-Cognitive version, which involves a cognitive dual task, is described below in Motor Cognitive variables.

## Motor cognitive

In the TUG-Cognitive assessment, participants complete TUG task while also counting backward by threes from a randomly selected number between 20 and 100, paying equal attention to walking and counting [44]. The TUG percent time change is calculated as the time (sec) for TUG-Cognitive minus the time needed to complete simple TUG, divided by simple TUG and multiplied by 100.

The Body Position Spatial Task (BPST) incorporates spatial memory and navigational skills while maintaining posture. The examiner demonstrates (verbally and visually) a pattern of side, forward, and turning (in place) steps, which the examinee repeats. The patterns increase in number of steps and turns with each additional level. Scores obtained are the number of correctly remembered sequences (Correct Trials) and length of the longest sequence remembered correctly (Span) [45].

## Cognition

The Delis-Kaplan Executive Function System™ (D-KEFS™) Color Word Interference Test (CWIT) [46] measures executive function over four conditions: color naming, word reading, inhibition, and inhibition/switching [47] Scaled scores have been age-adjusted by normative performance by age group per the Delis Kaplan Executive Function System manual. Corsi Blocks assesses short-term and working memory using nonverbal analog and consists of a

board containing nine cubes at fixed, pseudorandom positions. The blocks are labeled with numbers only visible to the experimenter. The experimenter taps several blocks, after which the participant attempts to tap this block sequence in the reverse order as that presented. The block sequences gradually increase in length, and the scores obtained are the number of correctly remembered sequences (Correct Trials) and the length of the longest sequence that was remembered correctly (Span) [48]. The Tower of London (ToL) assesses organization and planning ability, an aspect of executive function [49]. The administrator presents a card depicting a specific arrangement and the participants move five rings of varying sizes on three pegs to match the arrangement. The number of moves and the time it takes to complete the task are recorded.

### Psychosocial

The Beck Depression Inventory-II (BDI-II) is a self-report of the behavioral manifestation of depression [50] over the previous two weeks. Higher scores indicate more depressive symptoms.

### Data analysis

Secondary analyses were performed on baseline data from longitudinal studies conducted in our lab between 2011–2019 [51–54]. Data normality was tested using the Shapiro test and skewness and kurtosis, using Q-Q plots (not presented). Sample sizes for each variable are indicated in the tables. Missing data were observed and recorded. Some data were not available for lack of compliance (patient refused) and for operator error (approximately 1–3% of data points). Descriptive analyses describe demographic variables. Differences between groups were determined with independent t-tests for continuous variables and Chi-square test for categorical variables. Multiple linear regression analyses compared the mean of outcome variables between men and women with adjustments for age (Model 1), age and housing, (Model 2) and age, housing, PD duration and number of falls in the past year (Model 3). Age, PD duration and number of falls were chosen to be covariates because of their relevance to rehabilitation outcomes that could have been influenced by gender. For example, in the general older adult population, injurious falls occur more often in older women than men [55]. We also examined BMI as a covariate as well but it did not alter the findings beyond Models 1, 2 and 3. The $P$ value significance level was set at $< 0.05$.

Some data were not available for some variables because of assessments not having been administered for these participants due to scheduling or other barriers, participant refusal (rare), data not having been adequately or accurately captured (rare), because the assessment was not administered to a particular cohort, or because some measures, e.g., the MDS-UPDRS were more widely and popularly used later in the course of the study years, whereas the UPDRS was used earlier in the study.

### Results

All data were normally distributed. Demographic and clinical characteristics for each gender group are presented in Table 1. The gender distribution within the current sample (63.81% men and 36.19% women) is similar to previous studies [9, 25, 30]. Participants were 20 percent non-white, had mild-moderate PD (stages I-III) and the equivalent of a bachelor's degree in education attainment. Differences were noted between men and women on housing status, and this was therefore used in the model. Men were less likely to live in senior housing than women ($p < 0.01$).

**Table 2. Differences between women and men on disease severity, symptoms and psychosocial function.**

| | | Women | | Men | Un-adj P | Model 1 (Age-adjusted) | | | Model 2 (Age & Housing Adjusted) | | | Model 3 (Full Model) | | |
|---|---|---|---|---|---|---|---|---|---|---|---|---|---|---|
| | N | M ± SD | N | M ± SD | | Beta (95% CI) | β^ | Adj P | Beta (95% CI) | β^ | Adj P | Beta (95% CI) | β^ | Adj P |
| **Disease Severity** | | | | | | | | | | | | | | |
| MDS-UPDRS | | | | | | | | | | | | | | |
| Part I | 52 | 10.71 ±7.27 | 90 | 13.44 ±7.64 | 0.039* | 2.55 (-0.1, 5.2) | 0.16 | 0.059 | 2.98 (0.3, 5.7) | 0.19 | 0.031* | 3.13 (0.5, 5.7) | 0.20 | 0.019* |
| Part II | 51 | 10.63 ±7.07 | 89 | 16.88 ±8.87 | <0.001** | 6.54 (3.6, 9.5) | 0.36 | <0.001** | 6.75 (3.7, 9.8) | 0.37 | <0.00** | 6.79 (3.9, 9.6) | 0.37 | <0.01** |
| Part III | 72 | 31.83 ±12.16 | 127 | 34.1 ±12.14 | 0.207 | 3.18 (-0.3, 6.7) | 0.13 | 0.075 | 3.74 (0.2, 7.3) | 0.15 | 0.041* | 4.05 (0.6, 7.5) | 0.16 | 0.022* |
| Part IV | 51 | 3.1±3.45 | 89 | 4.2±3.8 | 0.089 | 0.83 (-0.5, 2.1) | 0.11 | 0.206 | 0.84 (-0.5, 2.2) | 0.11 | 0.212 | 0.96 (-0.3, 2.2) | 0.12 | 0.133 |
| **Psychosocial** | | | | | | | | | | | | | | |
| PDQ-39 | | | | | | | | | | | | | | |
| Mobility | 68 | 20.71 ±21.56 | 122 | 25.16 ±22.13 | 0.182 | 4.37 (-2.3, 11) | 0.1 | 0.198 | 6.7 (0, 13.4) | 0.15 | 0.05 | 7.89 (1.5, 14.3) | 0.17 | 0.016* |
| ADL | 69 | 17.66 ±16.08 | 124 | 27.72 ±20.47 | <0.001** | 9.2 (3.5, 14.9) | 0.23 | 0.002** | 9.47 (3.6, 15.3) | 0.23 | 0.002** | 10.48 (4.8, 16.1) | 0.26 | <0.001** |
| Emotional Well Being | 69 | 18±20.27 | 123 | 23.07 ±18.98 | 0.084 | 4.11 (-1.6, 9.8) | 0.1 | 0.159 | 5.82 (0, 11.6) | 0.14 | 0.049* | 6.4 (0.6, 12.2) | 0.16 | 0.03* |
| Stigma | 69 | 15.88 ±19.23 | 123 | 14.68 ±18.95 | 0.677 | -2.48 (-8.1, 3.1) | -0.06 | 0.381 | -1.6 (-7.3, 4.1) | -0.04 | 0.58 | -0.27 (-5.9, 5.3) | -0.01 | 0.925 |
| Social Support | 68 | 15.84 ±19.85 | 123 | 17.17 ±26.99 | 0.697 | 0.1 (-7.3, 7.5) | 0 | 0.98 | 0.9 (-6.6, 8.4) | 0.02 | 0.812 | 1.28 (-6.3, 8.8) | 0.02 | 0.739 |
| Cognitive Impairment | 69 | 23.34 ±17.51 | 124 | 28.73 ±20.58 | 0.068 | 5.2 (-0.7, 11.1) | 0.13 | 0.083 | 5.16 (-0.9, 11.2) | 0.13 | 0.094 | 6.23 (0.2, 12.3) | 0.15 | 0.044* |
| Communication | 69 | 18.63 ±20.27 | 124 | 26.41 ±20.71 | 0.013* | 7.68 (1.5, 13.9) | 0.18 | 0.015* | 7.81 (1.5, 14.1) | 0.18 | 0.015* | 9.45 (3.4, 15.5) | 0.22 | 0.002** |
| Bodily Discomfort | 69 | 33.33 ±23.44 | 124 | 30.31 ±20.81 | 0.356 | -3.73 (-10.2, 2.8) | -0.08 | 0.258 | -3.06 (-9.7, 3.6) | -0.07 | 0.367 | -2.19 (-8.9, 4.5) | -0.05 | 0.521 |
| Summary Index | 68 | 20.62 ±14.41 | 124 | 24.05 ±14.4 | 0.116 | 2.73 (-1.5, 7) | 0.09 | 0.209 | 3.63 (-0.7, 8) | 0.12 | 0.099 | 4.64 (0.4, 8.9) | 0.15 | 0.032* |
| Beck Depression Inventory-II (/63) | 62 | 10.53 ±7.86 | 110 | 13.74 ±8.51 | 0.016* | 2.71 (0.1, 5.3) | 0.16 | 0.041* | 3.4 (0.8, 6) | 0.19 | 0.011* | 3.51 (0.9, 6.1) | 0.20 | 0.009** |

Performance on measures of disease severity and symptoms and psychosocial function is presented.

^β: Standardized beta. Model 1 is age-adjusted, Model 2 is age and housing type adjusted and Model 3 is age, housing type, PD duration and number of falls in the past year adjusted; a PDQ-39 Question (Q) Scoring: 5-point Likert Scale (0–4); Mobility (MOB) = 100*((sum([Q1-Q10]))/(4*10)); ADL = 100*((sum([Q11-Q16]))/(4*6)); Emotional Wellbeing (EWB) = 100*((sum([Q17-Q22]))/(4*6)); Stigma (STIG) = 100*((sum([Q23-Q26]))/(4*4)); Social Support (SS) = 100*((sum([Q27-Q29]))/(4*3)); Cognitive Impairment (CI) = 100*((sum([Q30-Q33]))/(4*4)); Communication (COMM) = 100*((sum([Q34-Q36]))/(4*3)); Bodily Discomfort (BD) = 100*((sum([Q37-Q39]))/(4*3)); Summary Index = (sum([MOB], [ADL], [EWB], [STIG], [SS], [CI], [COMM], [BD]))/8. Lower scores are better for the PDQ-39 and the MDS-UPDRS.

*p < 0.05.

**p < 0.01.

Mean gender differences for disease severity & symptoms, motor, cognitive, motor-cognitive, and psychosocial measures are presented in Tables 2 and 3 along with adjusted models evaluating differences between groups.

Men exhibited worse disease severity and symptoms, including greater non-motor impact on experiences of daily living (MDS-UPDRS-I; $p < 0.05$ in Models 2 & 3), greater motor

**Table 3. Differences between women and men on motor, cognitive, and motor-cognitive function.**

| | Women | | Men | | Un-adj P | Model 1 (Age-adjusted) | | | Model 2 (Age & Housing Adjusted) | | | Model 3 (Full Model) | | |
|---|---|---|---|---|---|---|---|---|---|---|---|---|---|---|
| | N | M ± SD | N | M ± SD | | Beta (95% CI) | β^ | Adj P | Beta (95% CI) | β^ | Adj P | Beta (95% CI) | β^ | Adj P |
| **Motor** | | | | | | | | | | | | | | |
| Fullerton Balance (/40) | 50 | 25.32 ±8.8 | 89 | 27.87 ±7.45 | 0.072 | 1.74 (-0.7, 4.2) | 0.1 | 0.169 | 1.17 (-1.3, 3.7) | 0.07 | 0.355 | 0.72 (-1.7, 3.2) | 0.04 | 0.564 |
| Preferred Gait Speed (m/s) | 64 | 1.01 ±0.28 | 106 | 1.01 ±0.24 | 0.997 | -0.02 (-0.1, 0.1) | -0.03 | 0.697 | -0.04 (-0.1, 0) | -0.08 | 0.275 | -0.05 (-0.1, 0) | -0.09 | 0.234 |
| Backward Gait Speed (m/s) | 63 | 0.58 ±0.27 | 106 | 0.66±0.3 | 0.078 | 0.06 (0, 0.1) | 0.1 | 0.171 | 0.03 (-0.1, 0.1) | 0.06 | 0.44 | 0.03 (-0.1, 0.1) | 0.04 | 0.558 |
| Fast Gait Speed (m/s) | 64 | 1.31 ±0.36 | 106 | 1.45 ±0.38 | 0.019* | 0.12 (0, 0.2) | 0.15 | 0.045* | 0.08 (0, 0.2) | 0.1 | 0.167 | 0.08 (0, 0.2) | 0.1 | 0.202 |
| **Motor-Cognitive & Cognitive** | | | | | | | | | | | | | | |
| Timed Up and Go-Cognitive (s) | 70 | 18.29 ±20.09 | 120 | 14.65 ±8.41 | 0.153 | -2.67 (-6.8, 1.4) | -0.09 | 0.2 | -0.72 (-4.7, 3.3) | -0.02 | 0.722 | -0.29 (-4.3, 3.7) | -0.01 | 0.886 |
| TUG Pct. Time Change (100*C.-S./S.) (%) | 70 | 44.45 ±43.45 | 122 | 32.3 ±38.95 | 0.048* | -11.04 (-23.1, 1) | -0.13 | 0.072 | -9.34 (-21.7, 3) | -0.11 | 0.138 | -8.96 (-21.5, 3.6) | -0.11 | 0.161 |
| Serial 3 Correct Subtractions | 72 | 6.43 ±3.65 | 124 | 8.83 ±4.03 | <0.001** | 2.18 (1, 3.3) | 0.26 | <0.001** | 2.1 (0.9, 3.3) | 0.25 | <0.001** | 1.93 (0.8, 3.1) | 0.23 | 0.001** |
| Corsi Span | 72 | 4.21 ±1.11 | 125 | 4.52 ±1.42 | 0.09 | 0.27 (-0.1, 0.7) | 0.1 | 0.16 | 0.21 (-0.2, 0.6) | 0.08 | 0.292 | 0.19 (-0.2, 0.6) | 0.07 | 0.357 |
| Corsi Trials | 72 | 5.61 ±1.84 | 125 | 6.02 ±2.32 | 0.171 | 0.32 (-0.3, 0.9) | 0.07 | 0.302 | 0.18 (-0.4, 0.8) | 0.04 | 0.58 | 0.14 (-0.5, 0.8) | 0.03 | 0.672 |
| BPST Span | 68 | 3.57 ±0.87 | 124 | 3.83 ±1.04 | 0.085 | 0.19 (-0.1, 0.5) | 0.09 | 0.204 | 0.09 (-0.2, 0.4) | 0.05 | 0.525 | 0.07 (-0.2, 0.4) | 0.03 | 0.637 |
| BPST Trials | 68 | 4.21 ±1.58 | 124 | 4.48 ±1.67 | 0.263 | 0.12 (-0.4, 0.6) | 0.03 | 0.624 | -0.02 (-0.5, 0.4) | -0.01 | 0.916 | -0.09 (-0.6, 0.4) | -0.03 | 0.715 |
| CWIT-Inhibition Scaled Score | 44 | 10.09 ±3.44 | 70 | 8.91 ±3.76 | 0.095 | -1.12 (-2.5, 0.3) | -0.15 | 0.117 | -1.15 (-2.6, 0.3) | -0.15 | 0.117 | -1.35 (-2.8, 0.1) | -0.18 | 0.061 |
| CWIT-Inhibition/ Switching Scaled Score | 44 | 9.59±3.7 | 67 | 8.88 ±3.39 | 0.3 | -0.9 (-2.3, 0.5) | -0.12 | 0.197 | -1.05 (-2.4, 0.3) | -0.15 | 0.137 | -1.09 (-2.5, 0.3) | -0.15 | 0.117 |
| CWIT-Contrast Scaled Score Inhibition vs. Color Naming | 44 | 9.98 ±2.84 | 70 | 9.51±3.1 | 0.424 | -0.55 (-1.7, 0.6) | -0.09 | 0.347 | -0.79 (-2, 0.4) | -0.13 | 0.178 | -0.91 (-2.1, 0.3) | -0.15 | 0.125 |
| CWIT-Contrast Scaled Score Inhibition/ Switching vs. Inhibition | 44 | 9.2±2.78 | 67 | 9.72 ±2.98 | 0.366 | 0.4 (-0.7, 1.5) | 0.07 | 0.485 | 0.49 (-0.7, 1.6) | 0.08 | 0.405 | 0.53 (-0.6, 1.7) | 0.09 | 0.37 |
| Tower of London Total Achievement Score | 50 | 15.22± 4.22 | 90 | 15.58 ±5.37 | 0.685 | 0.03 (-1.7, 1.8) | 0 | 0.971 | -0.02 (-1.8, 1.7) | 0 | 0.982 | -0.16 (-1.9, 1.6) | -0.02 | 0.852 |
| Tower of London Time Ratio Scaled | 50 | 8.46 ±3.93 | 90 | 8.62 ±4.24 | 0.824 | -0.03 (-1.5, 1.4) | 0 | 0.972 | -0.01 (-1.5, 1.5) | 0 | 0.993 | 0 (-1.5, 1.5) | 0 | 0.995 |

(*Continued*)

**Table 3.** (Continued)

| | Women | | Men | | Un-adj *P* | Model 1 (Age-adjusted) | | | Model 2 (Age & Housing Adjusted) | | | Model 3 (Full Model) | | |
|---|---|---|---|---|---|---|---|---|---|---|---|---|---|---|
| | N | M ± SD | N | M ± SD | | Beta (95% CI) | β^ | Adj *P* | Beta (95% CI) | β^ | Adj *P* | Beta (95% CI) | β^ | Adj *P* |
| Tower of London Rule Violations | 50 | 2.82 ±5.31 | 90 | 3.27 ±4.51 | 0.6 | 0.8 (-0.9, 2.5) | 0.08 | 0.353 | 1.04 (-0.7, 2.8) | 0.1 | 0.231 | 1.01 (-0.7, 2.7) | 0.1 | 0.25 |

Performance on measures of motor, cognitive, and motor-cognitive function is presented.

^β: Standardized beta. Model 1 is age-adjusted, Model 2 is age and housing type adjusted and Model 3 is age, housing type, PD duration and number of falls in the past year adjusted TUG = Timed Up & Go; BPST = Body Position Spatial Task; CWIT = Color Word Interference Test; ToL = Tower of London; Lower scores are better for the PDQ-39 and the MDS-UPDRS.

*p < 0.05.

**p < 0.01

impact of PD on experiences of daily living (MDS-UPDRS-II; $p < 0.001$ in Models 1, 2 & 3), and greater motor signs of PD (MDS-UPDRS-III; $p < 0.05$ in Models 2 & 3). (Table 2)

On mobility measures, women exhibited slower gait speed than men ($p < .05$; Model 1). No gender differences were detected in self-selected gait speed ($p > 0.05$; adjusted β range: -0.03 — -0.09), backward gait speed ($p > 0.05$; adjusted β range: 0.04–0.1), and functional balance (as measured by FAB; $p > 0.5$; adjusted β range: 0.04–0.1). (Table 3)

For the cognitive and motor-cognitive measures, women performed worse on serial 3 subtractions ($p < 0.001$; Models 1, 2 & 3). (Table 3)

For the psychosocial and QoL measures, men exhibited worse scores for the PDQ-39 subscales of mobility ($p < 0.05$; Model 2), ADL ($p < 0.001$; Models 1, 2 & 3), emotional well-being ($p < 0.05$; Models 2 & 3), communication ($p < 0.05$ for Models 1 & 2; $p < 0.01$ for Model 3), cognitive impairment ($p < 0.05$; Model 3) and Summary Index Score ($p < 0.05$; Model 3). Men also exhibited more depressive symptoms ($p < 0.05$ for Models 1 & 2; $p < 0.01$ for Model 3; Table 2).

## Discussion

The current study aimed to examine and compare motor, motor-cognitive, cognitive and psychosocial function between men and women with diagnosed mild-moderate PD. Novel findings of this study demonstrate for the first time that in these patients residing in the Southeast of the United States (US), the impact of gender is most evident in PD-specific measures of motor symptoms, QoL and activities of daily living (ADL). Men experienced overall greater impact of the disease than women as reflected by the PDQ-39, a measure of HRQoL, motor signs and motor and non-motor experiences of daily living (e.g., speech, swallowing, hygiene, dressing, handwriting, mobility skills and cognition, apathy, sleep, pain, and depression). For non-PD measures of function, men experienced more depression, while women had slower fast gait speed and gave fewer correct answers during serial 3s subtractions. The most striking differences were noted in PD-specific measures of motor and psychosocial function, while the differences between genders in non-PD specific measures were quite minor in this sample.

### Gender differences in PD-related non-motor symptoms

Men in this sample exhibited greater impact of PD on non-motor experiences of daily living, per the MDS-UPDRS (parts I and II). Reports that used other instruments showed that women had more severe non-motor symptoms than men per the Non Motor Symptoms Scale (NMSS) [9], including sleep/fatigue, mood/apathy, constipation, restless legs, and pain, and

sexual dysfunction [56]. Two other studies reported that women had more severe symptoms of fatigue [9, 57, 58], light-headedness, fainting, lack of interest in surroundings, and lack of motivation, feelings of nervousness and sadness [9, 57]. Worse QoL could have been impacted by worse disease severity, greater motor and non-motor impact of PD on daily activities, and more depression, all experienced by men in our study. Worse PD related QoL in men aligns with other studies that also used PDQ-39: One study with 210 people with PD (61.4% men; *M* age: 69.1±10.8) from Australia found lower PD related QoL for men in the ADL, communication and cognition domains [22]. Other studies, however, reported worse PD related QoL in women, with some studies using the SF-12 questionnaire [23, 24, 30], and others using a validated translated version of the PDQ-39 [56]. Age differences may have played a minor role: in a study that analyzed 12 PD cohorts, the mean age ranged from 59.56 to 70.53 (males) and 59.51 to 70.14 (females) [21]. Cultural and social factors, such as perceiving "quality of life" could also serve as covariates explaining this opposing trend in non-motor symptoms. For example, the PDQ-39 components of social support and bodily discomfort are influenced by cultural and social norms, which may resonate more with men than with women in this sample. These potential explanations are in line with previous studies which highlighted the controversial role of gender in QoL in people with PD [56].

It is possible that the PD-specific measures are more attuned to issues that most affect men from this region, as opposed to the women. The disease is widely known to be slightly more prevalent in men, and while the measures were intended to be gender neutral, whether this holds true for different regions, with varying overall health status, social expectations and education levels, remains to be determined.

## Gender differences in PD-related motor symptoms

Finding worse scores on PD-related motor experiences of daily living (per the UPDRS-II) and PD motor signs (per the UPDRS-III) in men aligns with previous studies suggesting men have greater (i.e., worse) ADL [21, 59, 60] and motor [21, 22, 32, 61] symptoms severity. The lack of gender difference in self-reported indication of medication-related motor fluctuations, including dyskinesias (as measured by UPDRS-IV) does not align, however, with a previous report that found that females presented more dyskinesia than males [21]. In several studies, no differences between genders in scores on the UPDRS–Motor Examination (or the Movement Disorders Society revision (UPDRS-III), were shown [32, 56, 57, 59, 62–64]. Other work demonstrated women scored worse on the UPDRS-III [31], had more postural problems [32], and had worse ADL impairments [30]. Such inconsistency between research findings might be explained by between-studies differences in measures (e.g., the use of modified UPDRS-III and specific sub-scale scores (e.g., instability) or the use of P15-item Penn Parkinson's Disease Daily Activities Questionnaire for assessing cognition-related instrumental functional abilities [65], sample size [21, 30], participants' mean age [21, 31], ethnicity distribution [32], and geographical and cultural elements [31, 57]. On medication-related motor fluctuations (as measured by MDS-UPDRS-IV), another study also observed no gender difference [32].

Another possibility is that the men in the Southeastern region of the United States may be physically sicker than the women, which is true of the general population of older adults, given that women generally outlive men by 5 or more years [66]. This overall greater health burden may be reflected in higher symptom scores seen in these southern men.

## Gender differences in non-PD specific functional mobility

No gender differences were found in any of the non-PD specific functional mobility measures (besides gait speed). Slower fast gait speed in women, does not align with another study with

78 participants with PD that did not detect a gender difference in fast gait speed [67], possibly because of the use of normalized gait speeds (versus the use of non-normalized speeds in the current study) to calculate gender differences [67] and because these participants had longer disease duration (e.g., disease duration of 8.50±4.88 years versus 6.64±4.58 years in the current study). Fast gait speed demands maximal performance which may rely more of the ability of fast twitch muscles to generate power. Women tend to lose power in their lower limbs earlier than do men with the aging process [68]. The lack of gender difference in self-selected gait speed aligns with another study [67] while not aligning with other studies that detected slower gait speed in women over a longer distance (10-meters) [69]. Other studies assessing self-selected gait speed in people with PD could not be adequately compared to the current study due to significant differences in gait task characteristics and other methodological issues [70]. In sum, gender may affect gait speed, self-selected or fast, with women possibly walking slower during self-selected, due to average height differences, if data are not normalized and slower during fast gait, possibly because of the greater power involved.

The lack of gender differences (within adjusted models) in TUG-Cognitive and percent time change agree with a study that demonstrated no gender difference in gait speed interference (i.e., the relative change in dual-task walking speed in relation to walking speed only) [70].

## Gender differences in cognitive function

The only gender differences in cognitive performance noted was worse performance of women on serial 3s subtraction, a test of mental status and working memory. A search revealed no other literature that also examined this test between genders in PD. No gender differences were detected for global cognition and executive function (measured by MoCA, CWIT, Tower of London), confirming some work [56, 57, 71, 72] while contradicting other work, including a study with 490 people with PD (*M* age: 67.9±9.3; 62.4% men) [32] and a study that analyzed longitudinal data from 12 PD cohorts [21], which found MoCA scores to be significantly lower in men. The latter study did not find a significant differences between genders in The Mini-Mental State Examination [21], a screening tool for cognitive impairment, considered to be less challenging than the MOCA and which was not assessed in the current study. The lack of gender differences in visuo-spatial function per the Corsi blocks, and the whole body spatial cognition test, BPST in the current study, agrees with previous reports that measured visuospatial ability in people with PD using mental rotation of objects [73], problem-solving tasks [74], and Constructional apraxia [75]. Other reports have suggested women had worse visuospatial function, including as measured by the Corsi block test in a study with 306 people with PD [75] and by the Wechsler Adult Intelligence Scale-Chinese Revision graphic arrangement test in a study with 311 Chinese people with PD (*M* age: 60.85±11.35, 55.31% males) [72]. Further, a study with 31 people with PD (51.61% females, *M* age: 60/62.8) found that women with PD performed worse (i.e., had greater veering) while walking straight with eyes closed, a task used to measure spatial navigation [76]. This measure of veering shares some similarity with the BPST [40] used here, as both involve physically spatially navigating. Interestingly the Cognitive Impairment subdomain of the subjective PDQ-39, a measure of HRQoL, indicated men reported their HRQoL was more impacted by cognitive impairment than women, which adds to the PD-specific measures that were differentiated by gender.

## Gender differences in non-PD specific non-motor signs

A novel finding derived from this inquiry was that this sample of men reported greater depression (BDI-II; Models 1, 2 & 3) than women. Depression and anxiety were found by numerous

studies to be more prevalent among women [7, 9, 25, 30, 56, 57, 71, 77, 78]. A study that analyzed baseline data from a multi-center study conducted by the National Institute of Health with 1121 (62.5%) males and 615 females (*M* age not reported) did not find a gender difference on the BDI-II measures, although the cohort was overall less depressed for both genders (BDI-II scores: women: 7.1±5.7; men: 6.8±5.5) [60] compared to the current study. Another study [9] used item 12 on the NMSS for assessing depression in 950 people with PD (*M* age: 56.43±10.78; 62.63% men) and did not detect gender differences. Other studies who found women were more depressed used different measures, e.g., the Brief Symptom Inventory-18 [25] and the Montgomery-Asberg Depression Rating Scale [56], which could explain discrepancies in trends. Again, the unique status of southern United States men, who are comparatively sicker than other regions, may lead to the greater depression scores in men.

In sum, this study appears to support the research that suggests a potential protective impact of estrogens, higher dopaminergic levels, and more intact presynaptic system in women, which others have suggested [16]. The impact of gender is especially important and pronounced in the PD-specific measures in this cohort.

## Disentangling the literature with conflicting findings

Discrepancies between other studies' findings and ours could be explained by between-studies differences in participants' age, disease duration, race and ethnicity, and differences in cultural-related aspects. This study had 78.3% white participants, while other studies were mostly white [32] or included patients with more severe disease [71], whereas the current study H&Y range was by and large stages I-III, with a very few participants in stage IV. Further, the use of differing measures of different constructs could also affect results and their interpretations.

Findings at odds with the current study findings could originate from differences in scales used to measure symptoms, participants' age, sample size, percentage of participants taking anti-parkinsonian medications and spectrum of PD stages, disease duration [56], and social-cultural factors differing between cultures (e.g., social interactions, community support, and self-conceptualizations of QoL). Caution should be taken when considering gender findings within current literature, given: first, substantial differences among studies in operational definitions, constructs, and outcome measures, especially those used for assessing motor functioning [9, 32], QoL [56], and cognition [27, 56], and even PD-related measures. Second, participants' demographics, such as age [25, 56], ethnicity, and geography seem to impact gender differences in PD and thus should be thoroughly investigated and compared in future studies. Third, higher incidence of non-respondents among women, specifically relevant for epidemiological studies, has been suggested to affect research findings [79], and must be considered in the planning and design of future studies.

## Limitations

This study has several limitations. More men than women participated in this study. Although like other studies [16, 80], the potential reasons for lesser participation include underdiagnosis of women, and women potentially having demographic/social situations that lead to their inability to participate in a study, e.g., no caregiver to support them through the study. Importantly, women were recruited fairly in this study from Movement Disorders clinics, from support groups, and from local organizations that support people with PD. The lack of long-term follow-up prevents us from gaining knowledge regarding progression of the disease related to gender. Further, the current sample derives from a specific geographic area (the southeastern

US) and the findings cannot be generalized to the worldwide PD population or perhaps all regions of the US. The study lacks biomarker or neurophysiological measures, which would strengthen the rigor of the work. The neuropsychological battery lacks measures of language, and memory, which are crucial to examine for gender differences in a future study. The current study includes a single psychosocial measure of mood (BDI-II) while other neuropsychiatric symptoms (not considered in the current study) have been reported with gender differences in the related literature. In some variables, there were missing data, which may have influenced results in currently unknown ways. Future studies should also recruit from more non-white cohorts to be representative of all peoples. This study's categorization of Black, White and Other was an imperfect solution to ethnic/racial characterization of these participants. We included this information because there may be factors related to race that if better understood could lead to better outcomes for people of all races. All limitations should be addressed in future studies.

## Conclusion

A strength of this study is the wide battery of clinically relevant measures of motor and non-motor functions in PD that are used by many clinicians, giving the study more clinical relevance for neurorehabilitation approaches. In summary, this study adds to knowledge on the influence of gender on PD by suggesting that for individuals with mild-moderate PD, at least among this sample residing the Southeastern US, men are more impacted by burden of the disease on both QoL and self-reported motor and non-motor experiences of daily living as measured by PD-specific instruments. The current study also did not find statistically significant gender differences in functional balance (per the challenging FAB) and self-preferred and backward gait measures, as well as on several tests of executive function and visuospatial cognition. While this study does not provide the definitive story regarding functional measures and the genders, these effects were relatively robust. Except for the BDI-II, the differences between genders are most strongly noted in the PD-specific measures (MDS-UPDRS and PDQ-39). This trend was not expected, and the root causes of these differences are not fully understood. Possibly greater impact of PD exists for the men in this sample, as well as a more complex effect of gender and its various facets (e.g., physiological, psychological, and social) on the disease [26]. Another possibility is that because of the testing structure and the way test items are phrased or presented in PD-specific assessments, they are more likely to elicit differences between the genders. The current study provides behavioral knowledge which includes both self-reported as well as rater-reported measures that could promote a better understanding of the impact of PD on both genders. Further, this knowledge could promote the development of personalized gender-sensitive PD management policies and gender-sensitive assessment and rehabilitative protocols to be used in both research and clinical settings, including the development of gender-sensitive measures for assessing PD status, progression, and effectiveness of therapeutic interventions in this population. Implementing such knowledge in future research could assist overcoming some barriers toward PD optimal care for both men and women. The findings of this study should be further used, along with other literature, toward developing gender-targeted rehabilitation policies and guidelines. Such a process holds potential for fine tuning PD pharmaceutical and non-pharmaceutical therapies.

## Supporting information

**S1 Data.**
(CSV)

## Acknowledgments

We thank the volunteers and participants for their time and effort devoted to this study. The content is solely the responsibility of the authors and does not necessarily represent the official views of the National Institutes of Health.

## Author Contributions

**Conceptualization:** Ariyana Bozorg, Ariel R. Hart, Madeleine E. Hackney.

**Data curation:** Allison A. Bay, Nicole Schindler, Ariel R. Hart, Madeleine E. Hackney.

**Formal analysis:** Madeleine E. Hackney.

**Funding acquisition:** Madeleine E. Hackney.

**Investigation:** Amit Abraham, Liang Ni, Ariyana Bozorg.

**Methodology:** Madeleine E. Hackney.

**Project administration:** Amit Abraham, Ariyana Bozorg, Ariel R. Hart, Madeleine E. Hackney.

**Writing – original draft:** Amit Abraham, Allison A. Bay, Nicole Schindler, Eeshani Singh, Ella Leeth.

**Writing – review & editing:** Amit Abraham, Allison A. Bay, Madeleine E. Hackney.

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
