## [Decision Letter · Decision Letter 0]

4 Mar 2022

PONE-D-21-40790Gender Differences in Motor and Non-Motor Symptoms in Individuals with Mild-Moderate Parkinson’s DiseasePLOS ONE

Dear Dr. Hackney,

Thank you for submitting your manuscript to PLOS ONE. After careful consideration, we feel that it has merit but does not fully meet PLOS ONE’s publication criteria as it currently stands. Therefore, we invite you to submit a revised version of the manuscript that addresses the points raised during the review process.

We look forward to receiving your revised manuscript.

Kind regards,

Karsten Witt

Academic Editor

PLOS ONE

Journal Requirements:

a) Did participants provide their written or verbal informed consent to participate in this study?

Reviewers' comments:

Reviewer's Responses to Questions

**Comments to the Author**

1. Is the manuscript technically sound, and do the data support the conclusions?

Reviewer #1: Partly

Reviewer #2: Partly

2. Has the statistical analysis been performed appropriately and rigorously? 

Reviewer #1: Yes

Reviewer #2: Yes

3. Have the authors made all data underlying the findings in their manuscript fully available?

Reviewer #1: Yes

Reviewer #2: Yes

4. Is the manuscript presented in an intelligible fashion and written in standard English?

Reviewer #1: Yes

Reviewer #2: Yes

5. Review Comments to the Author

Reviewer #1: The authors present a cross-sectional retrospective study of gender differences in motor and non motor functions in PD patients living in the Southeast of the USA.

The novelty and relevance of their findings need to be emphasized in the discussion, with particular regard to the variables of rehabilitative interest. The paper by Iwaki et al (Mov Disord 2021, 36-106-117), that analysed gender differences in the largest sample of PD patients so far, should be discussed.

Some information on the physiopathological basis of gender differences in PD would be also advisable.

Minor comments:

Introduction, page 1: Ref 14 seems to support two opposite statements, please clarify.

In the methods, please clarify which are the motor-cognitive variables used.

Reviewer #2: In this study, authors investigated clinical gender differences in a cohort of idiopathic Parkinson's disease (PD) from the southern United States. Clinical motor, cognitive and psychosocial gender differences emerged mainly favoring women as compared to men. The sample size is quite large and the clinical battery is wide including many PD-specific measures that authors described adequately in the methods. Many clinical gender differences here reported in part confirm existing literature and otherwise suggest the need of investigating sociocultural factors depending on geographic areas as important determinants of gender differences in PD. Although no biomarkers are investigated, the topic is relevant, clinically applicable, and supports personalized medicine.

The study is well conducted, thus there are only few comments with the intention to improve the paper.

1. In the introduction (page 4, lines 89-94), the authors report many factors which can explain clinical gender differences. Among them, estrogens and other reproductive factors should be mentioned with adequate references because of their important influences on dopaminergic systems and related motor and non-motor symptoms.

2. Throughout all the manuscript, the authors refer to “mild-moderate PD”. However, it is not clear how they defined it since in the methods (Participants section) they reported the usual clinical criteria for PD. Despite it is not explicit, we supposed the mild-moderate definition is because of the H&Y stage. Please clarify this point.

3. The psychosocial measures here only include depression scale, other neuropsychiatric symptoms (not considered here) have been reported with gender differences in the related literature. This is a missing point that needs to be addressed in the discussion.

4. The reason to choice the number of falls in the past year as a covariate in the analyses is not clear.

5. According to the first comment, when authors discuss the motor gender differences that benefit women (totally aligned with the literature), they should also discuss the possible underlying mechanisms, such as the estrogens-induced neuroprotection.

6. In the discussion, the authors mainly compared their results to the previous evidence providing pointless details. Whereas they lack a discussion of their results in terms of biological differences, social differences, etc., a benefit from avoiding detailed comparisons with other secondary aspects and favoring a deep explanation and critical discussion is mandatory.

7. The relevance of sociocultural factors depending on geographic areas for PD gender differences is a crucial point that deserves a better discussion. The authors merely report this point as a possible explanation of findings not totally aligned with previous reports in comparable cohorts from other countries. Please provide a detailed explanation on how sociocultural and geographic factors might influence gender differen

6. PLOS authors have the option to publish the peer review history of their article (what does this mean?). If published, this will include your full peer review and any attached files.

Reviewer #1: No

Reviewer #2: No

---

## [Author Response · Author response to Decision Letter 0]

25 May 2022

Dear Dr. Witt, 

We are grateful to have the opportunity to revise this manuscript for consideration of publication in PLOS One. We have carefully considered all the reviewers’ helpful critiques and have revised the manuscript accordingly. We strongly believe the manuscript is much improved as a result. We look forward to hearing from you soon. 

Sincerely, 

Madeleine Hackney, Amit Abraham and co-authors. 

PONE-D-21-40790 

Gender Differences in Motor and Non-Motor Symptoms in Individuals with Mild-Moderate Parkinson’s Disease 

PLOS ONE 

Dear Dr. Hackney, 

Thank you for submitting your manuscript to PLOS ONE. After careful consideration, we feel that it has merit but does not fully meet PLOS ONE’s publication criteria as it currently stands. Therefore, we invite you to submit a revised version of the manuscript that addresses the points raised during the review process. 

We look forward to receiving your revised manuscript. 

Kind regards, 

Karsten Witt 

Academic Editor 

PLOS ONE 

Journal Requirements: 

RESPONSE: We have ensured that our manuscript meets PLOS ONE’s style requirements. 

a) Did participants provide their written or verbal informed consent to participate in this study? 

RESPONSE: We have clarified that all participants gave written informed consent (Methods/first paragraph). 

RESPONSE: Consent was written and this point has been clarified in Methods. 

RESPONSE: We have clarified and corroborated the information provided in Funding Information and Financial Disclosure. 

RESPONSE: We have ensured that the correct grant numbers are provided. 

RESPONSE: Thank you, we have uploaded the minimal anonymized data set necessary to replicate our study findings in a Supplemental Section. Therefore, all data are in the manuscript and supporting files 

RESPONSE: We have deleted the duplicative ethics statements. 

Reviewers' comments: 

Reviewer's Responses to Questions 

Comments to the Author 

1. Is the manuscript technically sound, and do the data support the conclusions? 

Reviewer #1: Partly 

Reviewer #2: Partly 

2. Has the statistical analysis been performed appropriately and rigorously? 

Reviewer #1: Yes 

Reviewer #2: Yes 

3. Have the authors made all data underlying the findings in their manuscript fully available? 

Reviewer #1: Yes 

Reviewer #2: Yes 

4. Is the manuscript presented in an intelligible fashion and written in standard English? 

Reviewer #1: Yes 

Reviewer #2: Yes 

5. Review Comments to the Author 

Reviewer #1: The authors present a cross-sectional retrospective study of gender differences in motor and non motor functions in PD patients living in the Southeast of the USA. 

The novelty and relevance of their findings need to be emphasized in the discussion, with particular regard to the variables of rehabilitative interest. 

RESPONSE: Thank you. We have emphasized the novelty and relevance of the findings in the Discussion, (1st paragraph). The new text reads: “Novel findings of this study demonstrate for the first time that for patients from a large metro area in the southeastern US, the impact of gender is most evident in PD-specific measures of motor symptoms, QOL and activities of daily living. … Overall, the most striking differences were noted in PD-specific measures of motor and psychosocial function, while the differences between genders in non-PD specific measures were minor in this sample.” 

The paper by Iwaki et al (Mov Disord 2021, 36-106-117), that analysed gender differences in the largest sample of PD patients so far, should be discussed. 

RESPONSE: Thank you. We have now included and addressed the paper by Iwaki et al. We referred to this paper and its findings in the Introduction (3rd paragraph) and Discussion (2nd, 3rd, 6th paragraphs). 

Some information on the physiopathological basis of gender differences in PD would be also advisable. 

RESPONSE: We agree with the reviewer that presenting information on the physiopathological basis of gender differences in important. Please find a new section in the Introduction (3rd and 4th paragraphs) that discusses these factors in depth. Thank you for the suggestion! 

Minor comments: 

Introduction, page 1: Ref 14 seems to support two opposite statements, please clarify. 

RESPONSE: Thank you. We have corrected the sentence. The new text reads: “some studies found certain aspects (activities of daily living (ADL), cognition and communication) of PD-related QOL to be rated lesser in men,13,14….” 

In the methods, please clarify which are the motor-cognitive variables used. 

RESPONSE: We have clarified the Motor Cognitive variables in Methods with a separate section. 

Reviewer #2: In this study, authors investigated clinical gender differences in a cohort of idiopathic Parkinson's disease (PD) from the southern United States. Clinical motor, cognitive and psychosocial gender differences emerged mainly favoring women as compared to men. The sample size is quite large and the clinical battery is wide including many PD-specific measures that authors described adequately in the methods. Many clinical gender differences here reported in part confirm existing literature and otherwise suggest the need of investigating sociocultural factors depending on geographic areas as important determinants of gender differences in PD. Although no biomarkers are investigated, the topic is relevant, clinically applicable, and supports personalized medicine. 

RESPONSE: Thank you. 

The study is well conducted, thus there are only few comments with the intention to improve the paper. 

RESPONSE: Thank you. 

1. In the introduction (page 4, lines 89-94), the authors report many factors which can explain clinical gender differences. Among them, estrogens and other reproductive factors should be mentioned with adequate references because of their important influences on dopaminergic systems and related motor and non-motor symptoms. 

RESPONSE: We agree and have included information (see response to Reviewer 1 above) about the pathophysiological differences between genders, related to estrogens and other reproductive factors. (See Introduction (3rd and 4th paragraphs). We touch upon the important influence these factors have on dopaminergic systems and related motor and non-motor symptoms. 

2. Throughout all the manuscript, the authors refer to “mild-moderate PD”. However, it is not clear how they defined it since in the methods (Participants section) they reported the usual clinical criteria for PD. Despite it is not explicit, we supposed the mild-moderate definition is because of the H&Y stage. Please clarify this point. 

RESPONSE: The reviewer is correct, Mild-moderate PD refers to stages I-III in Parkinson’s disease, and our participants were by and large representative of these stages. We have clarified this point in several locations where “mild-moderate” is iterated. 

3. The psychosocial measures here only include depression scale, other neuropsychiatric symptoms (not considered here) have been reported with gender differences in the related literature. This is a missing point that needs to be addressed in the discussion. 

RESPONSE: We agree with the reviewer. We have added a referral to this point in the Limitations section. Added text reads: “The current study includes a single psychosocial measure of mood (BDI-II) while other neuropsychiatric symptoms (not considered in the current study) have been reported with gender differences in the related literature.” 

4. The reason to choice the number of falls in the past year as a covariate in the analyses is not clear. 

RESPONSE: Thank you. We have added text for our reasons for including the falls in the past year as a covariate in the analyses in the Analysis section of Methods. The reason (in addition to that already given) is “For example, falls occur more often in older women than older men in the general older adult population.” 

5. According to the first comment, when authors discuss the motor gender differences that benefit women (totally aligned with the literature), they should also discuss the possible underlying mechanisms, such as the estrogens-induced neuroprotection. 

RESPONSE: We agree and have added two substantials paragraph of text devoted to the issue of gender differences in motor performance and symptoms that benefit women and the underlying mechanisms (e.g., estrogen-induced neuroprotection) in the Introduction (See response to Reviewer 1 above) and mention it again in Discussion.. 

6. In the discussion, the authors mainly compared their results to the previous evidence providing pointless details. Whereas they lack a discussion of their results in terms of biological differences, social differences, etc., a benefit from avoiding detailed comparisons with other secondary aspects and favoring a deep explanation and critical discussion is mandatory. 

RESPONSE: We appreciate the critique and have endeavored to discuss more deeply the biological and social differences that likely contributed to the observed results. We have endeavored to reduce text that contains trivial details. We discuss sociocultural factors that may have affected our cohort in the Introduction (5th paragraph) and allude to it several times in the DISCUSSION. 

7. The relevance of sociocultural factors depending on geographic areas for PD gender differences is a crucial point that deserves a better discussion. The authors merely report this point as a possible explanation of findings not totally aligned with previous reports in comparable cohorts from other countries. Please provide a detailed explanation on how sociocultural and geographic factors might influence gender differen 

RESPONSE: Thank you. We agree with this important point and have included a paragraph addressing the sociocultural and geographic factors that might have influenced gender differences in this sample. The new text is found in Introduction (5th paragraph) to motivate the research and this is alluded to throughout in Discussion. 

THANK YOU FOR THE HELPFUL CRITIQUES.

---

## [Decision Letter · Decision Letter 1]

29 Jul 2022

Gender Differences in Motor and Non-Motor Symptoms in Individuals with Mild-Moderate Parkinson’s Disease

PONE-D-21-40790R1

Dear Dr. Hackney,

We’re pleased to inform you that your manuscript has been judged scientifically suitable for publication and will be formally accepted for publication once it meets all outstanding technical requirements.

Kind regards,

Karsten Witt

Academic Editor

PLOS ONE

Additional Editor Comments (optional):

Reviewers' comments:

Reviewer's Responses to Questions

**Comments to the Author**

1. If the authors have adequately addressed your comments raised in a previous round of review and you feel that this manuscript is now acceptable for publication, you may indicate that here to bypass the “Comments to the Author” section, enter your conflict of interest statement in the “Confidential to Editor” section, and submit your "Accept" recommendation.

Reviewer #1: All comments have been addressed

Reviewer #2: All comments have been addressed

2. Is the manuscript technically sound, and do the data support the conclusions?

Reviewer #1: Yes

Reviewer #2: Yes

3. Has the statistical analysis been performed appropriately and rigorously? 

Reviewer #1: Yes

Reviewer #2: Yes

4. Have the authors made all data underlying the findings in their manuscript fully available?

Reviewer #1: Yes

Reviewer #2: Yes

5. Is the manuscript presented in an intelligible fashion and written in standard English?

Reviewer #1: Yes

Reviewer #2: Yes

6. Review Comments to the Author

Reviewer #1: All my comments have been addressed. I have no further comments for the authors.

Reviewer #2: (No Response)

7. PLOS authors have the option to publish the peer review history of their article (what does this mean?). If published, this will include your full peer review and any attached files.

Reviewer #1: **Yes: **Maria Teresa Pellecchia

Reviewer #2: No

---

## [Editor Report · Acceptance letter]

7 Sep 2022

PONE-D-21-40790R1 

Gender Differences in Motor and Non-Motor Symptoms in Individuals with Mild-Moderate Parkinson’s Disease 

Dear Dr. Hackney:

I'm pleased to inform you that your manuscript has been deemed suitable for publication in PLOS ONE. Congratulations! Your manuscript is now with our production department. 

Kind regards, 

on behalf of

Dr. Karsten Witt 

Academic Editor

PLOS ONE